# Geographic Variations and the Associated Factors in Adherence to and Persistence with Adjuvant Hormonal Therapy for the Privately Insured women Aged 18–64 with Breast Cancer in Texas

Junghyun Kim [1,*,†], Man S. Kim [2,†], Suja S. Rajan [3], Xianglin L. Du [4], Luisa Franzini [5], Tae Gi Kim [6], Sharon H. Giordano [7] and Robert O. Morgan [3,*]

1  Department of Preventive Medicine, College of Medicine, Yonsei University, Seoul 03722, Republic of Korea
2  Clinical Research Institute, Kyung Hee University Hospital at Gangdong, School of Medicine, Kyung Hee University, Seoul 05278, Republic of Korea; manskim@khu.ac.kr
3  Department of Management, Policy & Community Health, School of Public Health, University of Texas Health Science Center at Houston, Houston, TX 77030, USA; suja.s.rajan@uth.tmc.edu
4  Department of Epidemiology, Human Genetics & Environmental Sciences, School of Public Health, University of Texas Health Science Center at Houston, Houston, TX 77030, USA; xianglin.l.du@uth.tmc.edu
5  Department of Health Services Administration, School of Public Health, University of Maryland, College Park, MD 20742, USA; franzini@umd.edu
6  Department of Ophthalmology, Kyung Hee University Hospital at Gangdong, Kyung Hee University College of Medicine, Seoul 05278, Republic of Korea; tk1213@khu.ac.kr
7  Department of Health Services Research, The University of Texas MD Anderson Cancer Center, Houston, TX 77030, USA; sgiordan@mdanderson.org
*  Correspondence: jkim511@yuhs.ac (J.K.); robert.o.morgan@uth.tmc.edu (R.O.M.)
†  These authors contributed equally to this work.

**Abstract:** The purpose of this study is to examine the geographical patterns of adjuvant hormonal therapy adherence and persistence and the associated factors in insured Texan women aged 18–64 with early breast cancer. A retrospective cohort study was conducted using 5-year claims data for the population insured by the Blue Cross Blue Shield of Texas (BCBSTX). Women diagnosed with early breast cancer who were taking tamoxifen or aromatase inhibitors (AIs) for adjuvant hormonal therapy with at least one prescription claim were identified. Adherence to adjuvant hormonal therapy and persistence with adjuvant hormonal therapy were calculated as outcome measures. Women without a gap between two consecutively dispensed prescriptions of at least 90 days were considered to be persistently taking the medications. Patient-level multivariate logistic regression models with repeated regional-level adjustments and a Cox proportional hazards model with mixed effects were used to determine the geographical variations and patient-, provider-, and area-level factors that were associated with adjuvant hormonal therapy adherence and persistence. Of the 938 women in the cohort, 627 (66.8%) initiated adjuvant hormonal therapy. Most of the smaller HRRs have significantly higher or lower rates of treatment adherence and persistence rates relative to the median regions. The use of AHT varies substantially from one geographical area to another, especially for adherence, with an approximately two-fold difference between the lowest and highest areas, and area-level factors were found to be significantly associated with the compliance of AHT. There are geographical variations in AHT adherence and persistence in Texas. Patient-level and area-level factors have significant associations explaining these patterns.

**Keywords:** geographic variation; adjuvant hormonal therapy; breast cancer; adherence; persistence

## 1. Introduction

Breast cancer is the most commonly diagnosed cancer and the second leading cause of death in women in Texas, representing 29.5% of all new malignant cancers diagnosed

in women [1]. An average of 2849 Texas women with breast cancer died annually from 2012–2016 [2]. The expected breast cancer deaths were 3415 in 2022, the total hospital charges for breast cancer hospitalizations were approximately $252 million, and the estimated total expenditure for breast cancer care was highest for physician care at about $29.9 million compared to inpatient or outpatient hospital care in 2019 [1].

Significant variations in breast cancer care, including non-adherence to hormonal therapy and/or early discontinuation of hormonal therapy, arise frequently and may impact survival [3–6]. Many patients fail to take the prescribed drug daily (non-adherence) or to continue taking medication over a long-term period (persistence of use). Lack of compliance with prescribed adjuvant hormonal medication frequently results in treatment failure [4–7]. Adjuvant systemic hormonal therapy is a crucial procedure for maximizing the benefits of treatment. Guideline-concordant optimal adherence to, and persistence with therapy increases the likelihood that patients with hormonally sensitive breast cancer can expect better outcomes.

This study examines rates of adjuvant hormonal therapy compliance for patients aged 18–64 with breast cancer across different regions of Texas. Some studies have addressed regional variation in breast cancer's initial treatment type [3,8–14]. However, no study has compared adjuvant hormonal therapy adherence and persistence rates to determine whether geographical variations are seen among regions. Further, the primary predictors for observed patterns of quality adjuvant hormonal therapy have not been determined. There are mixed findings among studies for various individual and clinical characteristics, with estimates of geographic variation and provider/area-level factors frequently missing. It is unclear whether similar factors affect hormonal therapy use among the younger population as they do among older patients [4,5,7,15–20].

We compare actual endocrine therapy compliance rates by Hospital Referral Regions (HRR) across Texas. Texas is a particularly useful state in which to examine regional variation because it is large and diverse demographically and geographically. It is the second-most populous U.S. state and is estimated to have 29.95 million people in 2022 [21]. The state has several large urban and extensive rural areas and an ethnically diverse population. Some regions have access to large teaching hospitals, while many do not. We have access to private insurance claims data, which enables us to reliably determine endocrine therapy and follow the compliance of the therapy for younger patients over time and across all the regions in Texas.

The purpose of this study is to explore the geographic variation in HRR in Texas in terms of adherence to and persistence with adjuvant hormonal therapy use among privately insured women with breast cancer and its predictors.

## 2. Materials and Methods

### 2.1. Data

This study employed a retrospective population-based cohort, using enrollment and claims data for the population insured by the Blue Cross Blue Shield of Texas (BCBSTX) from 2008 to 2013. The dataset includes a claims file with all the claims (institutional, professional, and pharmacy claims) processed by the BCBSTX and a member enrollment file for approximately 5.6 million BCBSTX members from the years 2008 to 2013. This is approximately one-third of the private insurance population in Texas. The data were obtained through the University of Texas School of Public Health/BCBSTX Research Program in Payment Systems and Policy. The socioeconomic status of the areas where patients reside was obtained from the Census 2010 summary file 3 (SF3). Area-level characteristics were obtained from the 2012–2013 Health Resources and Services Administration's Area Resource File (ARF).

### 2.2. Study Cohort Description

The study cohort includes all women aged between 18 and 64 who were diagnosed with primary breast cancer (ICD-9 code, 174.x) and/or in situ breast cancer (ICD-9 code,

233.x) between 1 June 2008, and 31 December 2012. All women in the target population were continuously enrolled after the index date and followed for at least one and up to five years after their diagnosis.

Patients were included if they were enrolled in a preferred provider organization (PPO or PPO+) plan type, had drug benefits with the BCBSTX during the study period, and resided in a Texas Hospital Referral Region (HRR). We included patients who received at least one prescription for oral hormonal therapy such as tamoxifen, anastrozole, exemestane, or letrozole, after the index date and before disenrollment. We used the generic product index (GPI) code to identify all the maintenance adjuvant hormonal therapy (AHT) medications of interest from the pharmacy claims data.

*2.3. Outcome*

This study examined the rates of adherence and persistence of adjuvant hormonal therapy for treating breast cancer by Hospital Referral Regions in Texas. The International Society for Pharmacoeconomics and Outcomes Research (ISPOR) Work Group defined adherence as "the extent to which a patient acts in accordance with the prescribed interval and dose of a dosing regimen", and persistence as "the duration of time from initiation to discontinuation of therapy" [22]. Usually, adherence is described as the percent of doses taken as prescribed, and persistence is described as the number of days spent taking medication without exceeding a permissible gap [22].

Adherence was determined using a medication possession ratio (MPR), defined as the ratio of days covered by the amount of medication supplied over the total number of days during a defined period. It was considered to adhere to the AHT if the MPR was 80% or greater during a defined period by year [22–25]. Persistence is the length of time from initiation to discontinuation of therapy. Failure of persistence was defined as having a gap of at least 90 days between two consecutive prescriptions dispensed during the time from initiation to discontinuation of the AHT therapy. The use of 90 days (about 3 months) for defining persistence will be the primary gap definition in this study [22,24]. To account for potential variation in persistence, sensitivity analysis was conducted for the different gaps (60 and 180 days) in therapy, defining the persistence measure [5,16,26].

*2.4. Patient-Level, Provider-Level, and Area-Level Characteristics*

Patient-level variables include the age at diagnosis, year of diagnosis, whether the therapy was initiated within a year of breast cancer diagnosis, health care utilization (a total number of outpatient visits and inpatient days during the 6 months prior to breast cancer diagnosis and each subsequent year of therapy), Charlson comorbidity, which weights a range of comorbid conditions for a patient with a total of 22 conditions including heart disease, kidney disease, lung disease, AIDS, or cancer [27–29], as well as distance to care. Age was categorized into the following intervals: <40, 40–44, 45–49, 50–54, 55–59, and 60–64.

We also controlled for certain census tract-level indicators of patient socioeconomic status, which include the level of poverty in the area, level of educational attainment, and percent of the population that is non-white. Each of the four equal groups was divided according to the distribution of values for those variables. The percent of Hispanic and Latino Americans was distributed as a continuous variable. Zip code level variables were measured at the Zip Code Tabulation Area (ZCTA) level and assigned to the patient using the patient zip code and the zip code to ZCTA crosswalk available on the Dartmouth Atlas of Health Care website.

We determined the Hospital Referral Region (HRR) that each patient resided in. The Hospital Referral Region (HRR) represents the regional health care markets for their respective tertiary medical centers [30]. HRR unit analysis was conducted to describe geographic variation in adjuvant hormonal therapy adherence and persistence in Texas. The U.S. is divided into 306 HRRs, and 22 are in Texas. For MPR, the numerator was people who were adhering to the treatment, and the denominator was people who received AHT

in that HRR. In the regression analysis for adherence, the dependent variable was set equal to 1 if the adherence rate for the HRR was equal to or greater than 80% and 0 otherwise.

For provider-area characteristics (defined as county), the total number of hospitals, hospital admissions, and hospital beds as a proxy for hospital volume were estimated from the area resource file, with the data from most recent years were used. The county-level data from the ARF file was converted to HRR-level using a county-HRR crosswalk that weighs the population in each county. Area-level characteristics also include the number of oncology subspecialties (radiologists) in the ARF and the number of oncology providers in the area. The number of oncologists in the area was calculated over one calendar year for each hospital.

*2.5. Statistical Analysis*

Patients, providers, and area characteristics were summarized using percentages for categorical variables and means and standard deviations (SDs) for continuous variables. To examine geographic variation in adherence and persistence to adjuvant hormonal therapy, we reported the mean and median adherences and persistence rates for breast cancer adjuvant hormonal therapy in our study across Texas, as well as the lowest and highest compliance rates by HRR.

The adherence rates for each HRR were calculated by dividing the number of people who were already defined as adhering or not adhering to the treatment by the number of adherence counts in the HRR. We calculated the coefficient of variation and the index of variation to examine variation in adherence to guideline-recommended care among HRRs in Texas. We listed the name of the HRR associated with each reported rate so that we could look for similarities and differences in patterns of endocrine therapy care across regions in Texas.

To examine adherence to the hormonal therapy, we set the HRR with the median treatment rate for the sample as the excluded category in the regressions. We then tested for differences in AHT rates across HRRs; the dependent variable was set equal to 1 if the adherence rate for the HRR is equal to or greater than 80% and 0 otherwise, after adjusting for patient characteristics, census tract-level characteristics, provider characteristics, and area-level characteristics with the region of residence as a 22-level fixed effect. To describe variation by HRR in Texas in the persistence of adjuvant hormonal therapy use and to explore predictors of persistence, a Cox proportional hazards model with mixed effects was used to address regional variations of time to discontinuation without adjusting other covariates to be tested between each region and the median region for AHT persistence. In addition, sensitivity analyses with differing hormonal therapy duration gaps (60 days and 180 days) were conducted. All analysis was conducted using SAS version 9.4. and standard errors were computed using the cluster option to account for correlation in unobservables across HRRs.

**3. Results**

A total of 938 patients who had early breast cancer between 2008 and 2012 were identified. More than half of women had breast-conserving surgery (58.74%) and received radiation (97.12%). The majority of women were 45–59 years old at the time of diagnosis, lived in neighborhoods with a high school education, and were below the poverty line (Table 1). Of the 938 women with breast cancer, 627 (66.8%) initiated adjuvant hormonal therapy. Figure 1 addresses the steps of the study's inclusion and exclusion criteria.

Table 2 provided descriptive statistics on mean adjuvant hormonal therapy adherence and persistence rates by the HRRs, as well as information on the median, minimum, and maximum therapy adherence and persistence rates. We reported the number of people who received endocrine therapy in each HRR in the table.

The adherence rate is lowest in women from the HRR of San Angelo (54%), and it is highest in women from Victoria (100%). The median region for the adherence rate is Dallas (79%), and the mean rate for Texas was 79%. The median rate of persistence for a 90-day

gap in therapy is 81% in Dallas; the mean rate for Texas was 75%. The persistence rate is lowest in the HRR of Beaumont (50%) and highest in Bryan, San Angelo, Temple, and Victoria (100%). Figures 2 and 3 show the Texas maps with HRRs color-coded based on the adherence and persistence rates. The darker the color, the higher the compliance to the treatment.

**Table 1.** Descriptive statistics for the study cohort.

| Characteristic | N (%) |
|---|---|
| Adjuvant hormonal therapy | |
| Initiated therapy | 661 (66.2) |
| Tamoxifen only | 353 (53.4) |
| Aromatase inhibitor only | 308 (44.6) |
| Did not initiate therapy | 311 (33.8) |
| Cancer Treatment | |
| Breast conserving surgery | 573 (57.8) |
| Mastectomy | 329 (33.2) |
| Chemotherapy | 593 (59.8) |
| Radiation therapy | 964 (97.2) |
| Year of diagnosis | |
| 2008 | 238 (24.0) |
| 2009 | 234 (23.6) |
| 2010 | 210 (21.2) |
| 2011 | 179 (18.0) |
| 2012 | 131 (13.2) |
| Age at diagnosis | |
| <40 | 48 (4.8) |
| 40–44 | 87 (8.8) |
| 45–49 | 186 (18.8) |
| 50–54 | 247 (24.9) |
| 55–59 | 331 (33.3) |
| 60–64 | 93 (9.4) |
| Neighborhood, % non-white population | |
| <10% | 118 (11.9) |
| 10–24% | 481 (48.5) |
| 25–50% | 289 (29.1) |
| ≥50% | 104 (10.5) |
| Percent of the population aged 25 and older, without a high school education | |
| <25% | 775 (78.1) |
| ≥25% | 217 (21.9) |
| Neighborhood, % below poverty level | |
| <20% | 748 (75.4) |
| ≥20% | 244 (24.6) |
| Distance to health service facilities | |
| less than 5 mile | 287 (28.9) |
| 5–10 mile | 173 (17.4) |
| 10–35 mile | 312 (31.5) |
| 35–100 mile | 131 (13.2) |
| >100 mile | 89 (9.0) |
| Comorbidity at diagnosis, mean (SD) | 1.27 (2.0) |
| Health care utilization at diagnosis | |
| Outpatient visits in prior 6 months, mean (SD) | 1.74 (2.7) |
| Inpatient days in prior 6 months, mean (SD) | 1.47 (1.4) |

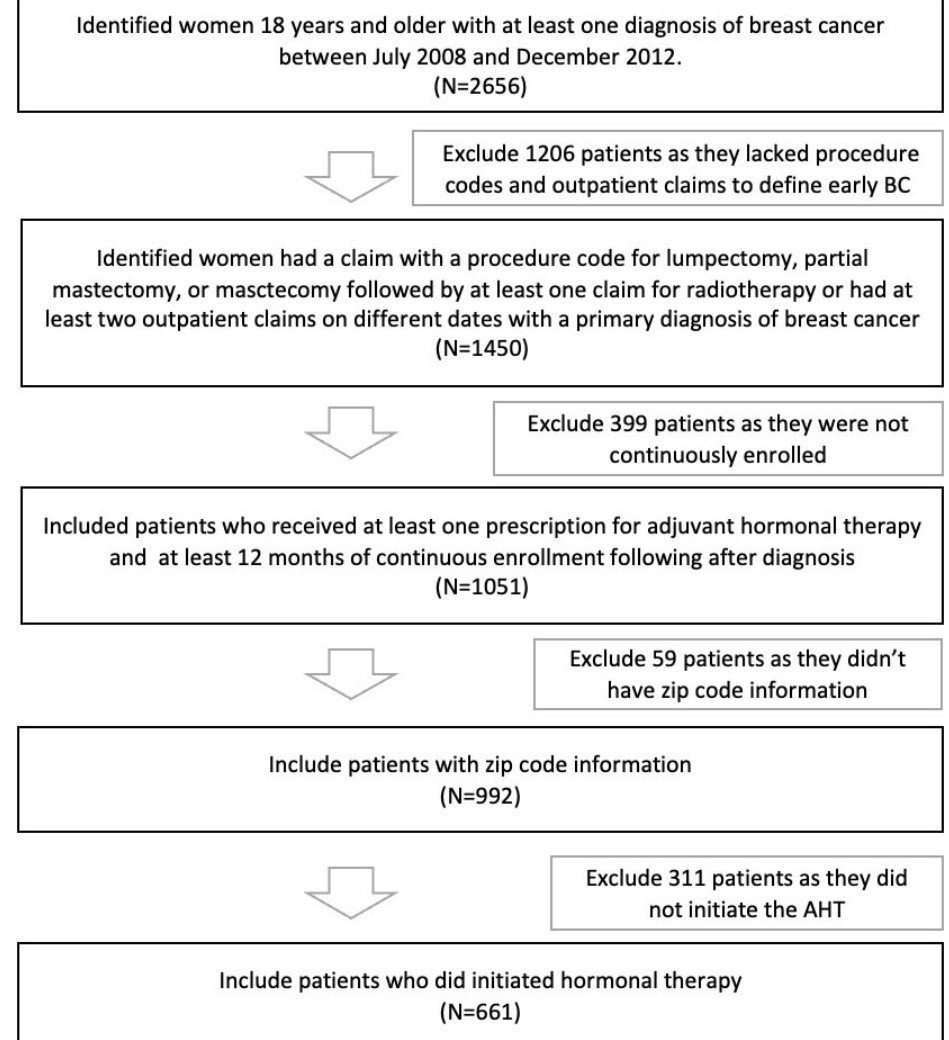

**Figure 1.** Steps of Inclusion/Exclusion Criteria.

Sensitivity analyses were conducted to see whether the 60-day or 180-day gaps in therapy affected the persistence rates across the regions. The median regions for the 60-day gap in therapy persistence rate and the 180-day gap in therapy persistence are Tyler (78%) and Lubbock (94%), respectively (Appendix A Figure A1). The Texas mapping with HRRs to address variations in the persistence of AHT across the regions can be found in Figures 2 and 3.

*3.1. Predictors of Adherence*

In the adjusted regression models for adherence to hormone therapy (Table 3), the patients who received BCS ($p < 0.011$) or chemotherapy ($p < 0.032$) were more likely to adhere to the AHT, whereas patients who resided in places where a larger non-white population lived ($p < 0.0001$), which had higher rates of poverty ($p < 0.014$), and more Hispanics or Latinos ($p < 0.009$) were less likely to adhere to the treatment. The number of outpatient visits ($p < 0.035$) and days of inpatient stay ($p < 0.04$) were associated with AHT adherence positively and negatively, respectively. The number of oncologists in the area ($p < 0.0001$) was a significant factor in adherence, and patients who initiated the hormonal therapy within a year of diagnosis as recommended ($p < 0.0001$) were more likely to adhere to the hormone therapy.

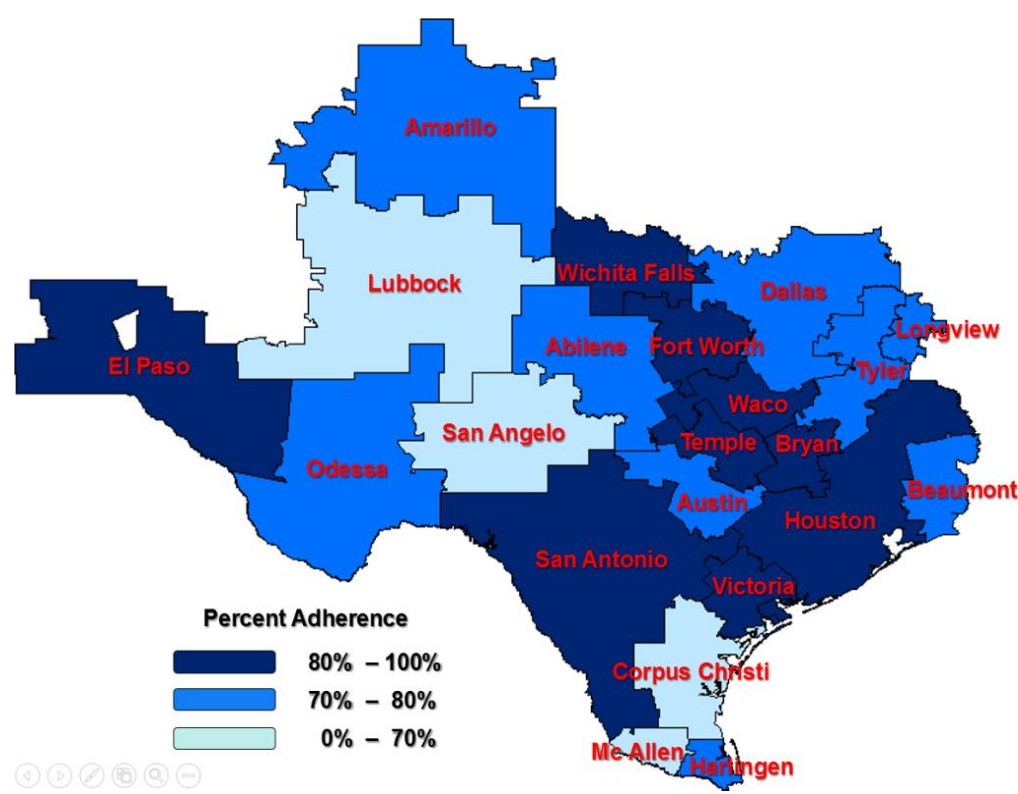

**Figure 2.** Adherence among Texas Hospital Referral Regions (HRRs); HRRs outside of Texas were not considered in the study.

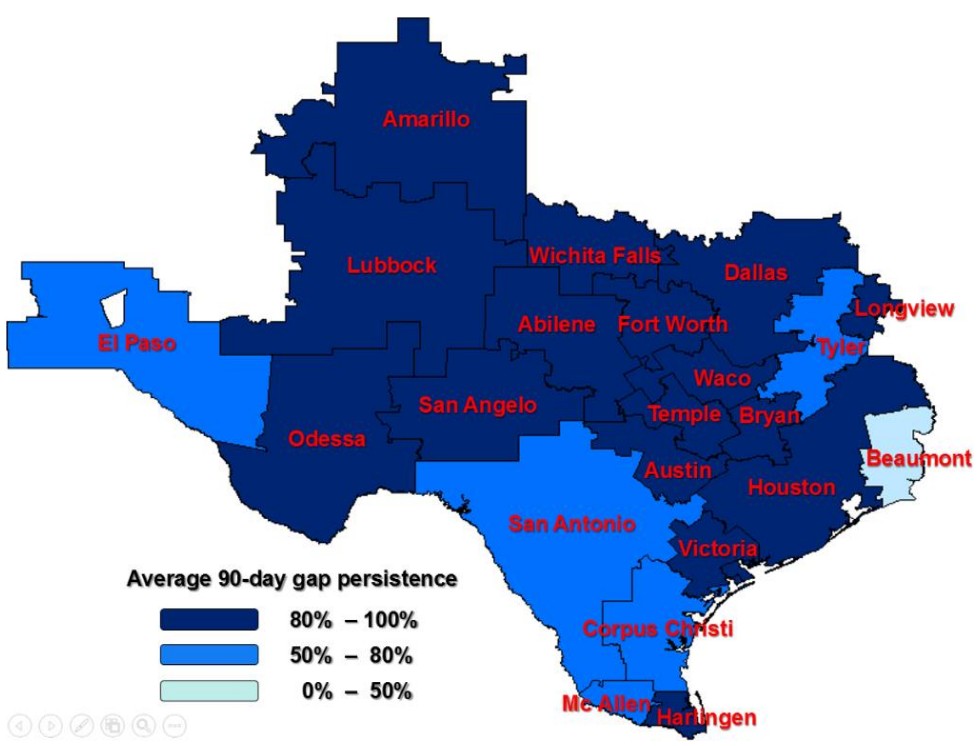

**Figure 3.** A 90-day persistence among Texas Hospital Referral Regions (HRRs); HRRs outside of Texas were not considered in the study.

**Table 2.** Mean AHT adherence and persistence rates by HRR.

| | N | Adherence by HRR * | | 90-Day Gap Persistence by HRR ** | |
|---|---|---|---|---|---|
| Austin | 57 | 0.77 | | 0.86 | |
| Bryan | 10 | 0.94 | | 1 | **Maximum region** |
| El Paso | 13 | 0.82 | | 0.78 | |
| **Houston** | 165 | 0.81 | | 0.80 | |
| Beaumont | 9 | 0.73 | | 0.50 | **Minimum region** |
| **Dallas** | 169 | **0.79** | **Median region** | **0.81** | **Median region** |
| Fort Worth | 55 | 0.83 | | 0.80 | |
| Harlingen | 8 | 0.71 | | 0.88 | |
| McAllen | 12 | 0.66 | | 0.67 | |
| Odessa | 21 | 0.71 | | 0.90 | |
| San Antonio | 51 | 0.84 | | 0.73 | |
| Temple | 1 | 0.8 | | 1 | **Maximum region** |
| **Victoria** | 1 | **1** | **Maximum region** | **1** | **Maximum region** |
| **San Angelo** | 2 | **0.54** | **Minimum region** | **1** | **Maximum region** |
| Waco | 5 | 0.96 | | 0.80 | |
| Wichita Falls | 2 | 0.93 | | 1 | Maximum region |
| **Abilene** | 9 | 0.7 | | 0.89 | |
| Corpus Christi | 5 | 0.61 | | 0.6 | |
| Amarillo | 15 | 0.78 | | 0.80 | |
| Longview | 15 | 0.77 | | 0.87 | |
| Lubbock | 18 | 0.68 | | 0.89 | |
| Tyler | 18 | 0.76 | | 0.78 | |

\* Mean adherence of TX = 0.79; ** Mean persistence of TX = 0.75; Region of median, minimum, and maximum therapy adherence and persistence rates are provided.

### 3.2. Predictors of Persistence

In the adjusted models for persistence of adjuvant hormonal therapy (Table 3), patients who were diagnosed with breast cancer in a later year were more likely to continue the hormone therapy ($p < 0.005$); patients who were further away from the health care services from the provider where they had the index treatment ($p < 0.012$) and who had more outpatient visits ($p < 0.02$) were less likely to persist in the treatment. Patients who initiated hormonal therapy within a year of diagnosis, as recommended ($p < 0.04$), were less likely to experience more gaps in therapy. The number of hospital admissions ($p < 0.04$), the number of hospital beds ($p < 0.04$), and the number of oncology practices in the area ($p < 0.02$) were associated with less and more patients experiencing gaps in therapy, respectively.

### Regional Variation in Persistence

Two regions, Amarillo and Houston, showed statistically significant unadjusted differences compared to the median region of Dallas for a 90-day gap persistence in therapy (Table 4). In the adjusted regression, three regions, Amarillo, Harlingen, and Odessa, were statistically and significantly different from the median region of Dallas. Amarillo remained significant, and Harlingen and Odessa became significant in the adjusted model.

Since gaps in therapy of other durations (60-day and 180-day) have also been used in the literature, we ran sensitivity analyses to see if regional variations in persistence differed by the length of the gaps (see Appendix A Tables A1–A3). One region (Amarillo) was found to be significantly different compared to the median region for average 180-day gap persistence in therapy; significant differences in regional variation were found using a 60-day gap persistence in therapy. The Texas mapping with HRRs to address variations in the persistence of AHT across the regions can be found in Appendix A Figures A1 and A2.

**Table 3.** Predictors of adherence to and persistence of adjuvant hormonal therapy.

| | Adherence | Persistence of 90-Day Gap in Therapy |
|---|---|---|
| | OR (CI) | HR (CI) |
| Patient-level characteristics | | |
| Cancer treatment | | |
| Surgery (referent: no cancer-directed surgery) | | |
| Breast-conserving surgery | **2.04 (1.18, 3.54)** | 1.69 (0.94, 3.04) |
| Mastectomy | 1.30 (0.73, 2.33) | 1.42 (0.76, 2.67) |
| Chemotherapy | | |
| (referent: no chemotherapy) | | |
| Yes | **1.61 (1.04, 2.48)** | 1.17 (0.95, 1.45) |
| Radiation therapy | | |
| (referent: no radiation therapy) | | |
| Yes | 1.10 (0.20, 6.17) | **0.39 (0.15, 0.97)** |
| Year of diagnosis (referent: 2008) | | |
| 2009 | 1.53 (0.87, 2.71) | **0.48 (0.27, 0.85)** |
| 2010 | 1.18 (0.66, 2.09) | 1.11 (0.60, 2.05) |
| 2011 | 1.54 (0.86, 2.75) | 1.29 (0.67, 2.48) |
| 2012 | 1.54 (0.80, 2.98) | **2.54 (1.13, 5.74)** |
| Therapy initiation (referent: did not initiated ATH within 1 year of BC diagnosis) | | |
| Not-Initiated | **0.02 (0.009, 0.04) *** | **0.87 (0.53, 0.98)** |
| Age at diagnosis, (referent: <40 years) | | |
| 40–44 | 2.28 (0.81, 6.37) | 1.25 (0.46, 3.46) |
| 45–49 | 1.54 (0.60, 3.98) | 1.43 (0.56, 3.67) |
| 50–54 | 2.07 (0.82, 5.23) | 2.57 (0.99, 6.60) |
| 55–59 | 2.49 (0.99, 6.30) | 1.98 (0.78. 5.01) |
| 60–64 | 1.12 (0.39, 3.21) | **3.44 (1.13, 10.52)** |
| Neighborhood, % non-white (referent: <10%) | | |
| 10–24% | 0.56 (0.30, 1.02) | **0.79 (0.39, 1.62)** |
| 25–50% | **2.47 (1.27, 4.81)** | **1.41 (0.64, 3.11)** |
| ≥50% | 1.16 (0.57, 2.38) | 1.87 (0.80, 4.35) |
| Percent of Hispanic and Latino population | **0.57 (0.39, 0.84)** | 1.03 (0.84, 1.25) |
| Neighborhood, % education lower than high school (referent: first quartile) | | |
| Second quartile | 1.64 (0.79, 3.39) | 0.54 (0.22, 1.34) |
| Third quartile | 1.55 (0.67, 3.67) | **0.23 (0.08, 0.72)** |
| Fourth quartile | 1.48 (0.57, 3.86) | **0.21 (0.06, 0.72)** |
| Neighborhood, % below poverty level (referent: first quartile) | | |
| Second quartile | **0.34 (0.16, 0.70)** | 2.12 (0.85, 5.30) |
| Third quartile | **0.32 (0.13, 0.78)** | **3.06 (1.02, 9.14)** |
| Fourth quartile | 0.55 (0.21, 1.43) | 3.27 (0.91, 11.75) |
| Comorbidity | 0.92 (0.83, 1.02) | 0.98 (0.88, 1.09) |
| Outpatient visits | **1.13 (1.00, 1.26)** | **0.84 (0.72, 0.98)** |
| Inpatient visits | **0.85 (0.72, 0.997)** | 0.88 (0.75, 1.03) |
| Distance to health service facilities (referent: less than 5 miles) | | |
| 5–10 mile | 0.92 (0.83, 1.02) | 0.98 (0.88, 1.09) |
| 10–35 mile | **1.13 (1.00, 1.26)** | **0.84 (0.72, 0.98)** |
| 35–100 mile | **0.85 (0.72, 0.997)** | 0.88 (0.75, 1.03) |
| >100 mile | 0.92 (0.83, 1.02) | 0.98 (0.88, 1.09) |
| Provider-level characteristics | | |
| Total number of hospitals, 2012 | 1.02 (0.98, 1.06) | 0.93 (0.82, 1.05) |
| Hospital admissions, 2012 | 1.00 (0.99, 1.001) | **1.00 (.99, 1.001)** |
| Hospital beds, 2012 | 1.00 (1.00, 1.001) | **0.998 (0.995, 1.001)** |
| Area-level characteristics | | |
| Total Subspecialty, 2013 | 0.99 (0.996, 1.002) | 0.99 (0.998, 1.001) |
| Number of oncology providers in the area | **0.99 (0.984, 0.991) *** | **0.99 (0.992, 0.999)** |

Bold numbers indicate statistically significant results ($p < 0.05$); Bold and asterisk numbers indicate statistically significant results ($p < 0.001$).

**Table 4.** Unadjusted and adjusted odds of AHT persistence by HRR.

|  | N | 90-Day Gap Persistence | |
|---|---|---|---|
|  |  | Unadjusted | Adjusted |
| Abilene | 9 | 0.96 | 0.91 |
| Amarillo | 15 | **0.22 [a] (0.003) [b]** | **0.14 (0.02)** |
| Austin | 57 | 1.70 | 0.99 |
| Beaumont | 9 | 0.33 | 0.57 |
| Bryan | 10 | - | - |
| Corpus Christi | 5 | 0.33 | 0.55 |
| Dallas | 169 | Ref | Ref |
| El Paso | 13 | 0.80 | 0.87 |
| Fort Worth | 55 | 0.98 | 0.48 |
| Harlingen | 8 | 1.21 | **3.03 (0.005)** |
| Houston | 165 | **0.50 (0.003)** | 1.05 |
| Longview | 15 | 1.74 | 1.43 |
| Lubbock | 18 | 1.52 | 2.04 |
| McAllen | 12 | 0.48 | 1.15 |
| Odessa | 21 | 2.82 | **10.29 (0.02)** |
| San Angelo | 2 | - | - |
| San Antonio | 51 | 0.56 | 1.02 |
| Temple | 1 | - | - |
| Tyler | 18 | 0.76 | 0.33 |
| Victoria | 1 | - | - |
| Waco | 5 | 0.29 | 0.46 |
| Wichita Falls | 2 | - | - |

[a] Bolded coefficients have *p*-values less than 0.05. [b] Numbers in parentheses are *p*-values. For ease of reading, *p*-values greater than 0.05 are not reported.

## 4. Discussion

Patient compliance with AHT is essential to maximize its significant benefits in cancer outcomes for early breast cancer survivors; therefore, disparities in the AHT compliance may partly contribute to the disparities in breast cancer outcomes. Our findings explore regional variations in adherence to and persistence of AHT for those with early breast cancer, and examine what factors might have been associated with those patterns across Texan regions. Our findings can help provide more precisely targeted local information that may be used to improve adjuvant therapy use and cancer care in Texas.

We find some significant differences in AHT persistence rates across regions of Texas. Six regions out of 22 reported discontinuing the medication administration with less than 90-day gaps in therapy. 10 regions and 1 region out of 22 discontinued taking the medication with less than 60-day gaps and 180-day gaps in therapy, respectively. Most of the smaller HRRs have significantly higher or lower rates of treatment adherence and persistence rates relative to the median regions.

The use of AHT varies from one geographical area to another, especially in terms of adherence, with an approximately two-fold difference between the lowest and highest areas. Areas in which the compliance rate is among the lowest quintile should be the focus of policies or strategies to increase the use of recommended care.

Socioeconomic factors were found to be significantly associated with the compliance of AHT. For adherence, patients who reside in lower SES areas (the places where more of the non-white population lives, there are higher rates of poverty, a lower proportion of the population receives a high school education, and more Hispanics or Latinos reside) have lower probabilities of receiving AHT. These findings were consistent with other studies [31–37]. Clinical factors, including the patients who received BCS or chemotherapy and the larger number of outpatient visits, have a positive effect on adherence. These findings are consistent with prior studies [6,20,38,39]. For persistence, patients who live in areas where more non-white population resides and the further distance from the

residential area to the health care services from the provider are less likely to persist in the treatment.

For both adherence and persistence, patients who initiated the hormonal therapy within a year of diagnosis, as recommended, were more likely to adhere to and continue the hormone therapy. National guidelines and the American Society of Clinical Oncology (ASCO) recommend that women with non-metastatic breast cancer initiate adjuvant hormonal therapy within a year of diagnosis [40]. The initiation of AHT medications and ensuring the continuity of care after the initiation should be emphasized since these interventions would reduce further exacerbation and recurrence.

One of the other predictors affecting adherence and persistence was access to care. It was partially explained by the number of hospital admissions and hospital beds, which were provider-level characteristics, and the number of oncologists, which were area-level factors. However, the more considerable number of oncologists in the area did not mean higher adherence rates in this study. This could be due to the patient's role in deciding to fill out a prescription or have a follow-up prescription or follow-up visit. Our data can only capture the patients who refilled their prescriptions and had follow-up visits with their providers. We have no information on whether providers may emphasize the importance of taking the medications as prescribed and recommended. Therefore, we cannot test the hypothesis as to whether it is the patients who are primarily responsible for making the decision to get their prescription refilled or for taking their medication as recommended; however, the lack of continuous medication management services for cancer survivors or the lack of seamless care transition from specialty care to primary care in these underserved areas may also influence patient adherence to AET in the long run [41,42].

There are several predictors of adherence and persistence that may help healthcare professionals identify populations with a higher risk of AHT non-compliance. It must be understood that factors affecting adherence are varied and complex, and interpersonal aspects of medication adherence should be appropriately investigated to employ a multi-measure approach that applies mixed methods of pharmacy database analysis and validated qualitative measures. Future research would address the increased adherence and persistence with more recent data and how the patterns have remained consistent or changed to implicate policymaking to help the people at high risk in Texas.

Our study has several limitations that need to be considered. Since administrative claims data were used in this study, we did not have clinical information, such as hormone receptor status, to justify the appropriateness of the initiation of AHT. However, we applied a well-known algorithm that was specifically developed for claims data to identify incident breast cancer cases [43]. This algorithm has been approved for its better performance in examining breast cancer cases using claims data. Examining adherence using prescription claims assumes that patients are taking medications as often as they fill prescriptions. Although using pharmacy records is the most accurate and validated estimate of actual medication use in large populations over periods of time [44–46], future research should address whether patients are taking the medication continuously as prescribed for follow-up.

Finally, since our study population included only women aged less than 65 who are enrolled in private insurance in Texas, the results may not generalize to patients who have public or no coverage or who reside in other states. The data from 2008 to 2013 that we used may not reflect the most current situation in this population. However, no study has been conducted regarding the examination of AHT compliance patterns of younger, privately insured women in Texas, and regional variation in the propensity of cancer patients to receive treatment may persist in a younger population [21]. This study helps to fill that gap.

In summary, we found substantial variations in the rates of adherence and persistence with AHT for privately insured Texan women with early breast cancer. Patient factors such as socioeconomic status are significantly associated with complying with treatment. System-level strategies, such as oncologists' explicitly recommending to women their medication use and asking about barriers to compliance with the therapy, especially for

those who reside in AHT underuse regions, may improve adherence to and persistence with the AHT and finally reduce further recurrence in patients with breast cancer.

**Author Contributions:** Conceptualization, J.K. and M.S.K.; methodology, J.K. and S.S.R.; software, L.F.; validation, X.L.D., S.H.G. and R.O.M.; formal analysis, J.K.; investigation, R.O.M.; data curation, M.S.K.; writing—original draft preparation, J.K.; writing—review and editing, J.K., M.S.K., S.S.R., S.H.G., T.G.K. and R.O.M.; visualization, M.S.K.; supervision, J.K. and R.O.M.; project administration, L.F. and T.G.K. All authors have read and agreed to the published version of the manuscript.

**Funding:** This research was supported by the National Research Foundation of Korea (2022R1F1A 1073848) and Kyung Hee University (20220780).

**Institutional Review Board Statement:** The study was conducted in accordance with the Declaration of Helsinki and approved by the Institutional Review Board (or Ethics Committee, AAI10249781) of the University of Texas Health Science Center at Houston.

**Informed Consent Statement:** Not applicable.

**Data Availability Statement:** The Blue Cross Blue Shield of Texas (BCBSTX) database, which supports the conclusions of this article, is not publicly available. Information on Hospital Referral Regions in Texas can be obtained at www.dartmouthatlas.org. (accessed on 1 August 2016).

**Conflicts of Interest:** The authors declare no conflict of interest.

## Appendix A

**Table A1.** Mean AHT persistence by HRR, by length of treatment gap.

| | N | 60-Day Gap Persistence by HRR * | 180-Day Gap Persistence by HRR * |
|---|---|---|---|
| Hospital Referral Region | | | |
| Abilene | 9 | 0.89 | 0.89 |
| Amarillo | 15 | 0.80 | 0.80 |
| Austin | 57 | 0.75 | 0.95 |
| Beaumont | 9 | 0.56 | 0.89 |
| Bryan | 10 | 1 | 1 |
| Corpus Christi | 5 | 0.40 | 1 |
| **Dallas** | 169 | 0.72 | 0.89 |
| El Paso | 13 | 0.77 | 0.85 |
| Fort Worth | 55 | 0.70 | 0.89 |
| Harlingen | 8 | 0.88 | 1 |
| Houston | 165 | 0.70 | 0.91 |
| Longview | 15 | 0.87 | 1 |
| **Lubbock** | 18 | 0.83 | **0.94** |
| McAllen | 12 | 0.67 | 0.67 |
| Odessa | 21 | 0.81 | 1 |
| San Angelo | 2 | 1 | 1 |
| San Antonio | 51 | 0.65 | 0.86 |
| Temple | 1 | 0 | 1 |
| **Tyler** | 18 | **0.78** | 0.83 |
| Victoria | 1 | 1 | 1 |
| Waco | 5 | 0.80 | 1 |
| **Wichita Falls** | **2** | **1** | **1** |

* Mean of TX = 0.69;0.90; Bolded coefficients have *p*-values less than 0.05.

**Table A2.** Unadjusted and adjusted odds of receiving AHT by Hospital Referral Region (by different days of gaps in therapy).

| | N | 60-Day Gap Persistence | | 180-Day Gap Persistence | |
|---|---|---|---|---|---|
| | | Unadjusted | Adjusted | Unadjusted | Adjusted |
| Abilene | 9 | 1.00 | 2.07 | 0.32 | 0.20 |
| Amarillo | 15 | 0.35 | 0.34 | **0.09 [a] (0.006) [b]** | **0.03 (0.0003)** |
| Austin | 57 | 0.93 | 1.56 | 2.73 | 1.08 |
| Beaumont | 9 | 0.25 | 0.33 | 0.36 | 0.21 |
| Bryan | 10 | 0.01 | - | - | - |
| Corpus Christi | 5 | 0.22 | 0.79 | - | - |
| Dallas | 169 | 1.18 | 1.51 | 0.83 | 0.21 |
| El Paso | 13 | 0.96 | 3.99 | 0.55 | 0.73 |
| Fort Worth | 55 | 1.42 | 0.75 | 1.11 | 0.18 |
| Harlingen | 8 | 0.78 | 7.46 | - | - |
| Houston | 165 | 2.27 | 0.90 | 1.28 | 0.27 |
| Longview | 15 | 2.04 | 2.19 | - | - |
| Lubbock | 18 | 1.60 | 4.44 | 1 | |
| McAllen | 12 | 0.70 | 4.70 | 0.2 | 0.47 |
| Odessa | 21 | 1.17 | 3.98 | - | - |
| San Angelo | 2 | 0.01 | - | - | - |
| San Antonio | 51 | 0.44 | 1.63 | 0.56 | 0.60 |
| Temple | 1 | - | - | - | - |
| Tyler | 18 | 1 | - | 0.46 | 0.12 |
| Victoria | 1 | 0.01 | - | - | - |
| Waco | 5 | 2.62 | 0.42 | - | - |
| Wichita Falls | 2 | 0.01 | - | - | - |

[a] Bolded coefficients have *p*-values less than 0.05. [b] Numbers in parentheses are *p*-values. For ease of reading, *p*-values greater than 0.05 are not reported.

**Table A3.** Predictors of failure to continue with adjuvant hormonal therapy by length of the gap.

| Characteristic | Duration of Gap in Therapy | |
|---|---|---|
| | 60 Days | 180 Days |
| **Patient-level characteristics** | | |
| **Cancer Treatment (referent: no mastectomy no BCS no chemo no rad)** | | |
| Mastectomy | 0.91 (0.72, 1.16) | 0.89 (0.72, 1.10) |
| BCS | 1.04 (0.84, 1.3) | 1.003 (0.82, 1.22) |
| Chemotherapy | **1.3 (1.04, 1.62)** | 1.19 (0.82, 1.22) |
| **Year of diagnosis (referent: 2008)** | | |
| 2009 | **1.39 (1.02, 1.89)** | **1.37 (1.04, 1.80)** |
| 2010 | **2.36 (1.71, 3.25) *** | **2.46 (1.86, 3.27) *** |
| 2011 | **3.95 (2.85, 5.49) *** | **3.80 (2.82, 5.11) *** |
| 2012 | **6.49 (4.5, 9.36) *** | **6.67 (4.75, 9.35) *** |
| 2009 | **1.39 (1.02, 1.89)** | **1.37 (1.04, 1.80)** |
| **Therapy initiation (referent: did not initiate ATH within 1 year of BC diagnosis)** | | |
| Initiated | **0.49 (0.31, 0.78) *** | **0.29 (0.19, 0.44) *** |
| **Age at diagnosis, (referent: <40 years)** | | |
| 40–44 | **0.58 (0.34, 0.99)** | 0.69 (0.43, 1.11) |
| 45–49 | 0.69 (0.43, 1.1) | 0.75 (0.49, 1.14) |
| 50–54 | 0.74 (0.47, 1.18) | 0.69 (0.45, 1.03) |
| 55–59 | 0.77 (0.48. 1.23) | 0.77 (0.51. 1.17) |
| 60–64 | 0.8 (0.48, 1.34) | 0.76 (0.48, 1.22) |
| **Neighborhood, % non-white (referent: <10%)** | | |
| 10–24% | 1.38 (0.99, 1.89) | **1.37 (1.03, 1.82)** |
| 25–50% | 1.31 (0.89, 1.93) | 1.30 (0.91, 1.84) |
| ≥50% | 1.71 (1.03, 2.84) | 1.35 (0.86, 2.13) |
| 10–24% | 1.38 (0.99, 1.89) | **1.37 (1.03, 1.82)** |

**Table A3.** *Cont.*

| Characteristic | Duration of Gap in Therapy | |
| --- | --- | --- |
| | 60 Days | 180 Days |
| Percent of Hispanic and Latino populations | 0.94 (0.76, 1.16) | **1.01 (0.84, 1.22)** |
| Neighborhood, % education lower than high school (referent: <25%) | | |
| ≥25% | 0.86 (0.64, 1.16) | 0.88 (0.67, 1.16) |
| Neighborhood, % below poverty level (referent: <20%) | | |
| ≥20% | 1.21 (0.90, 1.62) | 1.12 (0.86, 1.48) |
| Comorbidity | **0.95 (0.90, 1.0)** | 0.96 (0.92, 1.006) |
| Outpatient visits | 0.99 (0.91, 1.09) | 1.01 (0.94, 1.09) |
| Inpatient visits | 0.98 (0.91, 1.07) | 0.93 (0.93, 1.07) |
| Distance to health service facilities (referent: less than 5 mile) | | |
| 5–10 mile | 0.67 (0.49, 0.91) | **0.71 (0.54, 0.94)** |
| 10–35 mile | 0.80 (0.62, 1.04) | 0.92 (0.72, 1.16) |
| 35–100 mile | 0.78 (0.56, 1.09) | 0.89 (0.66, 1.19) |
| >100 mile | 0.82 (0.57, 1.19) | 0.93 (0.66, 1.30) |
| Provider-level characteristics | | |
| Total number of hospitals, 2012 | 1.02 (0.98, 1.06) | 1.02 (0.98, 1.06) |
| Hospital admissions, 2012 | 1.0 (.99, 1.001) | 1.0 (0.99, 1.001) |
| Hospital beds, 2012 | 1.0 (1.00, 1.001) | 1.0 (1.00, 1.001) |
| Area-level characteristics | | |
| Total Subspecialty, 2013 | 0.99 (0.996, 1.002) | 0.99 (0.998, 1.001) |
| Number of oncology providers in area | 0.99 (0.997, 1.001) | 0.99 (0.998, 1.003) |

Bold numbers indicate statistically significant results ($p < 0.05$); Bold and asterisk numbers indicate statistically significant results ($p < 0.001$).

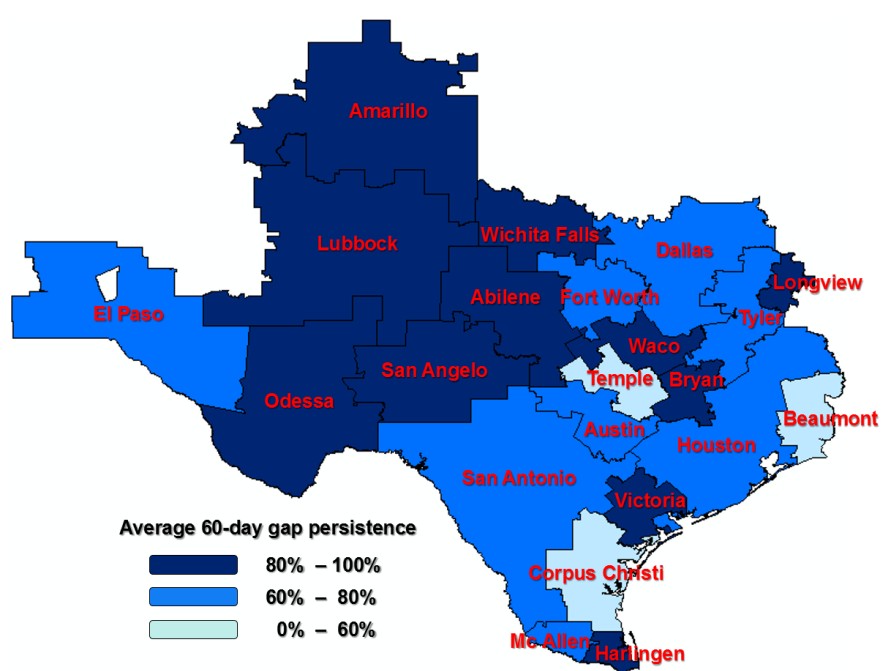

**Figure A1.** A 60-day persistence among Texas Hospital Referral Regions (HRRs); HRRs outside of Texas were not considered in the study.

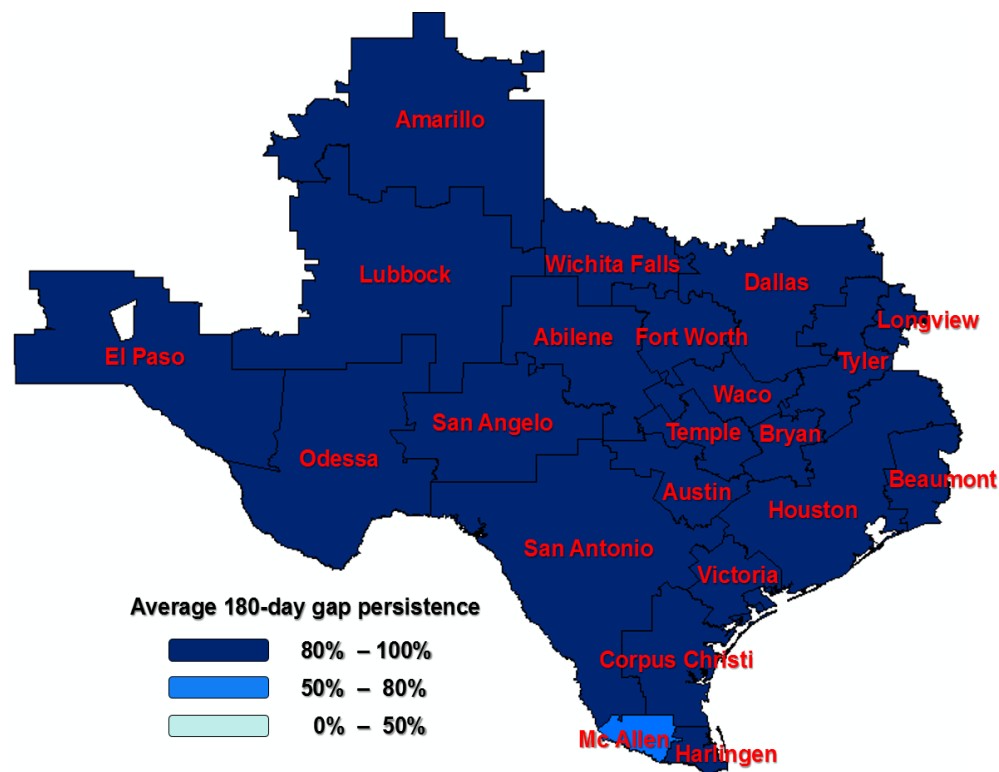

**Figure A2.** A 60-day persistence among Texas Hospital Referral Regions (HRRs); HRRs outside of Texas were not considered in the study.

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
