# Peer review of "Geographic Variations and the Associated Factors in Adherence to and Persistence with Adjuvant Hormonal Therapy for the Privately Insured women Aged 18–64 with Breast Cancer in Texas"

_curroncol, doi:10.3390/curroncol30040288_

Round 1
Reviewer 1 Report
This retrospective cohort study aimed “to explore the geographic variation by Hospital Referral Region in Texas in adherence to and persistence with adjuvant hormonal therapy use among privately insured women with breast cancer and its predictors”. The researchers analysed 5-year claims data for women aged 18-64 with early breast cancer privately insured in Texas. They found variations in the rates of adherence to and persistence with adjuvant hormonal therapy for women residing in different regions in Texas.
The study is potentially interesting, but it has many limitations, in particular about the generalizability of findings, because the authors focused the results on geographic variation, while they should focus on the socioeconomic differences underlying these geographic differences.
In particular, I would suggest to underline the role of the predictors, individual and area-based socio-economic ones, in adherence to and persistence with adjuvant hormonal therapy.
Moreover, the method section has to be considerably revised.
In particular, I have many concerns about the study, which are summarized below.
MATERIALS AND METHODS
· 2.3 Outcome (106-109): Adherence was determined using a medication possession rate (MPR), defined as the ratio of days covered by the amount of medication supplied, of 80% or greater during a defined period by year.
The authors could better define “MPR”, as a “proportion of days covered by the amount of medication supplied”, or as “the ratio of days covered by the amount of medication supplied over the total number of days during a defined period”.
· 2.4 Patient-level, Provider-level, and Area-level characteristics
The authors should classify in a clearer way the three levels described.
First, they should clearly distinguish the individual variables from the socio-economic ones measured at the “zip code level”. In particular, why do they consider “percent Hispanic and Latino population” as a patient-level variable? If the value of socio-economic variables is ecologically attributed to individuals, it must be must be made explicit in the text.
Moreover, it is not clear how they consider the variable “health care utilization”: as a count data (lines 118-120) or as a dummy variable (lines 130-131)?
· The adherence rates for each HRR were calculated by dividing the number of people who were already defined whether they were adhering or not adhering to the treatment by the number of adherences counts in the HRR.
The authors should clarify this definition, both for numerator and denominator.
· To examine regional variation of adherence to and persistence of the hormonal therapy, we used patient-level logistic regression models with region of residence as a 22-level fixed effect. Differences were tested between each region and the median region for each outcome (adherence and persistence of therapy).
The authors seem to describe the data as if they were hierarchically organized, describing variables as defined at Patient-level, Provider-level, and Area-level characteristics. They should better clarify this relevant issue, explaining better the data structure, because if so, probably the most appropriate analysis would be one that uses multilevel regression models.
· “The adherence by HRR were described in Table 2. For persistence by HRR, various lengths of treatment gaps were applied in Table 3 (lines 157-158)”.
Tables should not be listed in the Method section but in the Results one. Moreover, the description does not match with the effective results reported in Table 3.
RESULTS
· Maybe there is a typo in table 3, because the measure indicated for “Persistence of 90-day gap in therapy” is HR and not OR.
· The authors show in table 4 unadjusted and adjusted odds of receiving AHT by hospital referral region. It would seem that they calculated them considering the rates of AHT adherence and 90-day gap persistence compared to the reference values of Dallas HRR (line 217-218 and 237-239).
However, I do not understand how did they calculate these “odds” (or maybe Odds Ratios), considering the rate values that they show in table 2. For example, as the 90-day gap persistence rate of Amarillo was 0.80 and the Dallas one was 0.81, how did they obtain the unadjusted odd (OR) for Amarillo equal to 0.22? Can the authors explain this issue?
· The authors must clarify the regression models used to predict the results in Table 3 and Table 4. Are all the predictors showed in Table 3 used to predict the adjusted ORs showed in Table 4?
DISCUSSION
· In the Discussion, the authors should mainly underline the impact of socioeconomic factors on their findings. In fact, the differences that they observed among Texas HRRs are interesting, but are less so for readers outside of Texas or the United States.
· More in general, I think that to overcome the limitations in generalizability of their findings, the authors should focused the results that they showed and the discussion on the socioeconomic differences underlying geographic differences.
Author Response
"Please see the attachment."

Reviewer 2 Report
The aim of the study is to observe geographic variations in adherence to and persistence with adjuvant hormonal therapy for privately insured women aged 18-64 with breast cancer in Texas.
My biggest observation is that both the data and the references referred to the years prior to 2012. How is it possible that there are no more recent references and especially more recent data in the Texas data system?.... in these last 10 years many things have changed...
Moreover, I have some other concerns about the article.
The introduction doesn’t provide a strong rationale for the study.
At line 72 an estimate of the 2022 population is made with an unrelated reference from 2007.
The definitions of adherence and persistence are vague, in the methods section, the rationale reported for the use of these terms is unclear (e.g., lines 106-108 about adherence, are cited with 14 references).
Results
Figure 1 is unclear: in the main boxes included patients are reported and in the last two boxes those excluded); and in the boxes of excluded patients the reasons are not detailed.
Table 1 doesn’t have Title 1 and 2 and some % sums do not add up to 100%.
Table 2 presents an alphabetical order and is not easily understandable.
I suggest giving reasons for the aspects highlighted above to provide a solid rationale for the article, and give more attention to the methods and results section.
Author Response
"Please see the attachment.

Round 2
Reviewer 1 Report
The manuscript is improved in the present version compared to the original one. However, some concerns remain and I think that the authors should clarify in the text the following issues:
· It is written in the manuscript : “The adherence rates for each HRR were calculated by dividing the number of people who were already defined whether they were adhering or not adhering to the treatment by the number of adherences counts in the HRR”.
I suggested the authors to clarify the definition, both for numerator and denominator, and they did it in the answer to me, but not in the text of the manuscript.
I suggest them to rewrite the definition like they wrote in their answer: “Numerator is people who were adhering to the treatment and the denominator is people who received adjuvant hormonal therapy in that HRR. In regression analyses for adherence, the dependent variable was set equal to 1 if the adherence rate for the HRR is equal or greater an 80% and 0 otherwise”.
· In my previous revision I wrote: “The authors seem to describe the data as if they were hierarchically organized, describing variables as defined at Patient-level, Provider-level, and Area-level characteristics. They should better clarify this relevant issue, explaining better the data structure, because if so, probably the most appropriate analysis would be one that uses multilevel regression models”.
The answer was: “I agree with the opinion that multilevel regression models would fit in this study, however, the Cox proportional hazard model with mixed effects share many similarities with methods for the analysis for multilevel data with continuous, binary or count outcomes. The Cox proportional hazards model is enhanced through the incorporation of random effect terms to account for within-cluster homogeneity in outcomes (reference: Peter C. Austin. A Tutorial on Multilevel Survival Analysis: Methods, Models and Applications, International Statistical Review (2017), 85, 2, 185–203 doi:10.1111/insr.12214). Persistence was measured from initiation of the AHT to discontinuation, therefore survival analysis was applied in the study”.
This is right, but in the original version of their manuscript they did never declare the use of a Cox proportional hazard model with mixed effects. This omission also caused the misunderstanding that there might be a typo in Table 4 (HR instead of OR). Furthermore, it would seem that the authors used it for the outcome "persistence," but do not explain whether they took into account the hierarchical structure of the data for the outcome "adherence". They should clarify this issue.
· In my previous revision I wrote: “The authors show in table 4 unadjusted and adjusted odds of receiving AHT by hospital referral region. It would seem that they calculated them considering the rates of AHT adherence and 90-day gap persistence compared to the reference values of Dallas HRR (line 217-218 and 237-239). However, I do not understand how did they calculate these “odds” (or maybe Odds Ratios), considering the rate values that they show in table 2. For example, as the 90-day gap persistence rate of Amarillo was 0.80 and the Dallas one was 0.81, how did they obtain the unadjusted odd (OR) for Amarillo equal to 0.22? Can the authors explain this issue? The authors must clarify the regression models used to predict the results in Table 3 and Table 4. Are all the predictors showed in Table 3 used to predict the adjusted ORs showed in Table 4?
They answered: “As above mentioned, the Cox proportional hazards model with mixed effects was applied because of the characteristic (time to discontinuation) of persistence definition, and regression models were clarified in the method section, and all the predictors were used to predict the adjusted ORs and HRs showed in Table 4”.
As above mentioned, the authors did never declare the use of a “Cox proportional hazard model with mixed effects” in the original version of their manuscript. However, they do not clarify how they calculated the unadjusted Odds (or Odds Ratio?) presented. I think it would be important for the reader to understand this issue because it is difficult to understand how they got the presented values, considering the adherence rates in each HRR compared to the reference one of Dallas. The authors should explain this in a way that helps the reader interpret the results of their study.
Author Response
"Please see the attachment."

Reviewer 2 Report
The authors responded promptly to the comments made.
The tables have been adjusted as requested
Author Response
As the reviewer mentioned as follows:
"The authors responded promptly to the comments made.
The tables have been adjusted as requested"
No additional response is needed.
Thank you very much for your valuable suggestions!
Round 3
Reviewer 1 Report
· There’s been a misunderstanding, because the sentence “Numerator is people who were adhering to the treatment and the denominator is people who received adjuvant hormonal therapy in that HRR. In regression analyses for adherence, the dependent variable was set equal to 1 if the adherence rate for the HRR is equal or greater an 80% and 0 otherwise” should be written in the paragraph “2.4 Patient-level, Provider-level, and Area-level characteristics” (lines 166-168 of the 3rd version).
· However, my main concern remains the calculation of adherence and persistence rates (table 2) and the calculation of the “unadjusted and adjusted odds of receiving AHT by hospital referral region” (table 4). The authors showed in Figure 1 and in Table 1 that 627 patients initiated hormonal therapy. But in table 2 “mean AHT adherence and persistence rates by HRR” seem have been calculated the rates on 943 patients (the sum of the N showed in the table), a number that is not only different from 938 (all patients with a zip code, as shown in the previous box of Figure 1), but in any case it would also include the 311 patients “excluded as they did not initiated AHT”, according to Figure 1. Why did they calculate AHT adherence and persistence rates also including in the denominator those who had not started therapy? The authors should explain this issue. In table 4 “unadjusted and adjusted odds of receiving AHT by hospital referral region” were calculated on 661 patients (the sum of the N showed in the table), a number that is different from 627 (patients who initiated hormonal therapy). In any case, it would appear that, while adherence and persistence rates were calculated including patients who did not initiate therapy (table 2), odds ratios were calculated only on those who had initiated therapy (table 4). Therefore, it is difficult to understand how they obtained the values of unadjusted odds that one could calculate considering the adherence and persistence rates of each HRR compared to the reference rate of Dallas (assumed as reference). The authors should explain this issue.
Author Response
Please see the attachement

Round 4
Reviewer 1 Report
The authors corrected in numbers in Figure 1 and Table 2.
However, I still don't understand how the authors get the results of the unadjusted odds ratios in Table 4. For example, how is it possible that Austin's unadjusted odds ratio of ATH adherence I is 1.56, given that Austin's adherence rate is 0.77 and Dallas' (reference) 0.79; or how is it possible that Bryan's unadjusted odds ratio is 5.28, given that Bryan's adherence rate is 0.94 and Dallas' (reference) 0.79?
I asked several times to clarify this issue about unadjusted odds ratios calculation, without ever receiving answers that satisfy the question.
Author Response
Please see the attached.
Thank you very much for your suppot.
Best,
Junghyun
